# Response to marine cloud brightening in a multi-model ensemble

Camilla W. Stjern[1,2], Helene Muri[2], Lars Ahlm[3,4], Olivier Boucher[5], Jason N. S. Cole[6], Duoying Ji[7], Andy Jones[8], Jim Haywood[8], Ben Kravitz[9], Andrew Lenton[10], John. C. Moore[7,11,12], Ulrike Niemeier[13], Steven J. Phipps[14], Hauke Schmidt[13] Shingo Watanabe[15], Jón Egill Kristjánsson[2, †]

5   [1] CICERO Center for International Climate and Environmental Research Oslo, Oslo, Norway
[2] Department of Geosciences, University of Oslo, Oslo, Norway
[3] Department of Meteorology, Stockholm University, Stockholm, Sweden
[4] Bolin Centre for Climate Research, Stockholm University, Sweden
[5] Laboratoire de météorologie dynamique, Université Pierre et Marie Curie, Paris, France
10  [6] Canadian Centre for Climate Modelling and Analysis, Environment and Climate Change Canada, Victoria, Canada
7 College of Global Change and Earth System Science, Beijing Normal University, Beijing, China
[8] Met Office Hadley Centre, Exeter, UK
[9] Atmospheric Sciences and Global Change Division, Pacific Northwest National Laboratory, Richland, USA
[10] CSIRO Oceans and Atmosphere, Hobart, Australia
[11] Joint Center for Global Change Studies, Beijing, 100875, China
[12] Arctic Centre, University of Lapland, P.O. Box 122, 96101 Rovaniemi, Finland
[13] Max Planck Institute for Meteorology, Hamburg, Germany
[14] Institute for Marine and Antarctic Studies, University of Tasmania, Hobart, Australia
[15] Japan Agency for Marine-Earth Science and Technology, Yokohama, Japan
[†] Deceased, 14 August 2016

*Correspondence to:* Camilla W. Stjern (camilla.stjern@cicero.oslo.no)

**Abstract.** Here we show results from Earth System Model simulations from the marine cloud brightening experiment G4cdnc of the Geoengineering Model Intercomparison Project (GeoMIP). The nine contributing models prescribe a 50%
increase in the cloud droplet number concentration (CDNC) of low clouds over the global oceans in an experiment dubbed G4cdnc, with the purpose of counteracting the radiative forcing due to anthropogenic greenhouse gases under the RCP4.5 scenario. The model ensemble median effective radiative forcing (ERF) amounts to -1.9 Wm$^{-2}$, with a substantial inter-model spread of -0.6 to -2.5 Wm$^{-2}$. The large spread is partly related to the considerable differences in clouds and their representation between the models, with an underestimation of low clouds in several of the models. All models predict a
statistically significant temperature decrease with a median of (for years 2020-2069) -0.96 [-0.17 to -1.21] K relative to the RCP4.5 scenario, with particularly strong cooling over low-latitude continents. Globally averaged there is a weak but significant precipitation decrease of -2.35 [-0.57 to -2.96] % due to a colder climate, but at low latitudes there is a 1.19 % increase over land. This increase is part of a circulation change where a strong negative top-of-atmosphere (TOA) short-wave forcing over subtropical oceans, caused by increased albedo associated with the increasing CDNC, is compensated
by rising motion and positive TOA long-wave signals over adjacent land regions.

## 1    Introduction

The Paris Agreement of the United Nations Framework Convention on Climate Change (UNFCCC) 2015, 21[st] Conference
of Parties (UNFCCC, 2015) with its ambitious aims of limiting global warming to 2°C, if not 1.5°C to avoid dangerous climate change, has raised concerns over how to actually reach those targets. Climate engineering, also referred to as geoengineering, could be considered as part of a response portfolio to contribute to reach such targets. Climate engineering can be defined as the deliberate modification of the climate in order to alleviate negative effects of anthropogenic climate change (Sheperd, 2009). Marine cloud brightening (MCB) is one such technique (Latham, 1990), which falls into the

category of solar radiation management or albedo modification, and aims to cool the climate by increasing the amount of solar radiation reflected by the Earth.

The MCB method involves adding suitable cloud condensation nuclei (CCN), for instance sea salt, into the marine boundary layer. As existing cloud droplets tend to distribute themselves on the available nuclei, a larger number of CCN has the potential to enhance the cloud droplet number concentration (CDNC) in a cloud, which (given constant liquid water paths) can reduce the droplet sizes and therefore increase the cloud albedo (Twomey, 1974). In reality, however, the end result of adding CCN to the marine boundary layer is highly uncertain, as there are many processes involved – each of which has a number of dependencies and uncertainties. For instance, Alterskjær and Kristjánsson (2013) simulated sea salt seeding of marine clouds using the Norwegian Earth System Model (NorESM), and found that for seeding particles above or below given size thresholds, a strong competition effect ultimately led to a warming of the climate, contrary to the intention. Chen et al. (2012) studied observations of ship tracks and found that the magnitude and even the sign of the albedo response is dependent on the mesoscale cloud structure, the free tropospheric humidity, and cloud top height. Similarly, Wang et al. (2011), using the WRF model, found that the effectiveness of cloud albedo enhancement is strongly dependent on meteorological conditions and background aerosol concentrations. There is therefore a great need for more studies of the processes behind and possible effects of MCB.

Existing model studies of MCB (e.g. Alterskjær et al., 2013; Bala et al., 2011; Bower et al., 2006; Jones et al., 2009; Latham et al., 2014; Philip et al., 2009; Stuart et al., 2013; Wang et al., 2011) have shed some light on potential benefits, e.g. in terms of climate cooling, as well as drawbacks, e.g. hydrological cycle changes. For instance, while all of the above studies show that MCB "works" in terms of cooling climate, a spin-down of the hydrological cycle in the cooler climate leads to reduced precipitation in the global average, with potential detrimental effects to humans and vegetation (Muri et al., 2015). Jones et al. (2009) found a significant drying for the Amazon basin from increasing CDNC in marine stratocumulus decks. Bala et al. (2011) noted that differential cooling of oceans (where clouds were modified) versus land might result in circulation changes (seen also in Alterskjær et al. (2013)) involving sinking motion over oceans and rising motion over land, which therefore might experience an increase in precipitation. Thus, MCB may have very different regional effects. E.g., Latham et al. (2014) investigated possible beneficial regional effects of MCB, and finds that MCB might help stabilize the West Antarctic Ice sheet. However, comparison between studies has been difficult partly due to different experimental design. Also, in studies based on one or only a few models, results will be very dependent on the particular models' parametrizations. For instance, Connolly et al. (2014) suggested that the unintended warming found in Alterskjær and Kristjánsson (2013) for seeding of some particle sizes (as mentioned above) was likely to be an artifact of the cloud parametrization scheme in the model used. To alleviate some of these issues, the Geoengineering Model Intercomparison Project (GeoMIP) initiated a series of experiments where a number of models were to simulate MCB in a particular manner, with different degrees of complexity in the simulation design (Kravitz et al., 2013). The G1ocean-albedo experiment prescribes an increase in ocean albedo in the models at a rate intended to offset increasing global temperatures in response to a quadrupling of atmospheric $CO_2$ concentrations. The G4cdnc experiment investigated in this work simulates a 50 % increase in the CDNC of low marine clouds as described in more detail in Section 2. Finally, the G4sea-salt experiment involves an increase in sea salt emissions over tropical oceans at a rate intended to produce a radiative forcing of $-2.0\ Wm^{-2}$ under the RCP4.5 scenario. The three experiments have an increasing degree of complexity or realistic representation.

Less idealized experiments, such as sea salt injection simulations, will include more of the processes in play between geoengineering and climate impacts. Such experiments have taught us, for instance, that a substantial part of the cooling will originate from direct aerosol effects, and that the indirect cloud effects are just part of a number of responses of the climate system (Ahlm et al., 2017; Alterskjær et al., 2012; Partanen et al., 2012). In order to get a clearer understanding of the causes of the inter-model spread in cloud response, we therefore look more closely into the simpler experiment G4cdnc, where only perturbations to cloud simulations are made. The justification for performing more simplified simulations such as G4cdnc is that many models may participate, giving a more complete multi-model ensemble, but also that the spatial climate response in near surface air temperature and precipitation where CDNC are perturbed directly in a single fully coupled GCM is similar to that when sea-salt aerosol microphysics is included explicitly (Jones et al., 2009; Jones and Haywood, 2012). In this paper we provide an initial overview of the climate response in the G4cdnc experiment with a particular focus on the atmosphere. Our main goal is to determine the range of responses associated with such a forcing and furthermore to assess the extent to which the climate effects are dependent on the quantity, type, and location of clouds.

In the next section, we describe the G4cdnc experimental design, and the data analysis approach. Section 3 reviews the model climatologies, with a particular focus on differences in clouds. Results from the climate engineering experiment are presented in Section 4, Section 5 provides the discussion, whilst conclusions are drawn in Section 6.

## 2    Data and methods

### 2.1    The G4cdnc experiment design

The G4cdnc experiment uses the CMIP5 (Coupled Model Intercomparison Project 5) RCP4.5 (Taylor et al., 2012) scenario as its baseline. RCP4.5 is the middle-of-the-road scenario and assumes continued greenhouse gas emission increases from today's levels (albeit at reduced rate from a no-policy scenario like RCP8.5), followed by a decrease from year 2040 and stabilization by year 2100, at which time the anthropogenic radiative forcing amounts to 4.5 $Wm^{-2}$ above preindustrial levels (Meinshausen et al., 2011). The G4cdnc experiment starts the climate engineering in year 2020, and prescribes a 50% increase in the CDNC of marine low clouds. Marine low clouds are defined as clouds below 680 hPa over ocean grid boxes at all latitudes, except where sea ice is present. The experiment is run for 50 years, from 2020 to 2069, after which the cloud brightening is terminated, and the simulations are continued for a further 20 years (until 2089) to assess the termination effect. Nine CMIP5 models participated in the experiment. However, the termination period was only investigated in six of the models. See Table 1 for a list of participating models.

In the aforementioned G4sea-salt experiment, sea salt particles are injected close to the ocean surface to simulate the entire life-cycle from aerosol injections to climate effects (Ahlm et al., 2017). The G4cdnc experiment, however, simplifies the process and addresses the adjustment of CDNC without an actual increase in any cloud condensation nuclei (CCN) from sea spraying. Consequently, the resulting climate effects will only include aerosol-cloud interactions and will not include aerosol-radiation interactions (Boucher et al., 2013) or the climate system's adjustments to them. Note also that for BNU-ESM, the model design precluded a direct change in the CDNC. Therefore, instead of multiplying CDNC by 1.5 to obtain the 50% increase, they had to approximate the cloud seeding through a direct alteration of liquid droplet sizes (droplet radii are multiplied by the factor $1.5^{1/3}$), which means that its experiment set-up is slightly different from the other models.

## 2.2    Post-processing and analyses

Some details of the nine contributing models can be seen in Table 1. For each model, annual averages are first calculated from monthly mean model output and the ensemble means over all available realizations are then calculated, before the data is regridded to a horizontal grid corresponding to the average grid size of models; 2.1×2.7 latitude x longitude. When calculating differences between the G4cdnc experiment and the RCP4.5 scenario, we base our analyses on years 2020-2069. To assess whether the differences between G4cdnc and RCP4.5 are statistically significant, a two-sample non-parametric Kolmogorov-Smirnov (K-S) test (Conover, 1971) is used.

In our assessment, "low clouds" are quantified using a random overlap assumption, in which clouds in contiguous layers overlap in a random way (Tian and Curry, 1989), based on the cloud cover for all grid cells from 1000 hPa and up to 680 hPa. Please note that this estimate is based on monthly means, and these low cloud amounts will therefore not be equal to the low cloud fractions calculating during model runs (for models that does this); these numbers tend to be higher. However, they should give a fair representation of geographical distribution, and numbers are comparable between models. To estimate the effective radiative forcing of increasing CDNC by 50% in the different models, we use the method of Gregory et al. (2004), whereby the TOA radiative flux imbalance is regressed against the globally averaged surface air temperature change compared to the RCP4.5 simulations. To estimate the effective radiative forcing of increasing CDNC by 50% in the different models, we use the method of Gregory et al. (2004), whereby the global mean top of atmosphere radiative flux imbalance is regressed against the globally averaged surface air temperature change compared to the RCP4.5 simulations. Because of the time scales of the adjustments to the applied forcing, we follow Williams et al. (2008) and Gregory et al. (2004), and use annual means for the first decade, and decadal means thereafter; the period of 2030 - 2069. This allows more weight to the years when the rapid adjustments dominate, before the slow feedbacks have more impact in the later part of the simulation.

## 3    Modeled cloud climatologies

Earth system models have large differences in their treatment of clouds, as well as in their aerosol concentrations and climatologies. The 5[th] assessment report of the Intergovernmental Panel on Climate Change (IPCC),  similarly to the previous ones, stressed the important role of clouds and their parameterizations in contributing towards the inter-model spread in estimates of climate change (Stocker et al., 2013). As the G4cdnc experiment is designed so that changes are induced only to low clouds over ocean, we expect differences in cloud fields to be a particularly large source of uncertainty in our results. This section therefore provides a brief comparison of different climatological values between the models.

In Table 2, a selection of cloud related variables is given, based on global annual means from the first 20 years of the RCP4.5 simulation, to see how these values compare to present-day observations. Zonal mean cloud cover over the same period is shown in Fig. 1. The lowermost row in Table 2 shows values from observations. Looking at the observed cloud fraction of Fig. 1 (upper left panel), we see that low cloud amounts are particularly high near the poles, with also relatively high amounts extending towards the subtropics and mid-latitudes, especially in the Southern Hemisphere. Observations show a high fraction of low stratocumulus clouds around 30°S (Wood, 2012), an area that, given its distance from major

pollution sources, might be particularly susceptible to cloud seeding (Alterskjær et al., 2013; Jones et al., 2009; Partanen et al., 2012). It is therefore interesting to note that several models are not able to realistically reproduce these clouds.

Comparing individual model cloud fractions in Fig. 1 (lower three rows) to the observed cloud fraction, we see that few models compare well to the observed low-level cloud amounts. It is a well-known problem, commonly referred to as the 'too few, too bright' problem, that climate models tend to underestimate the amount of low clouds, while concurrently overestimating their optical thickness, and Nam et al. (2012) confirmed that this is true for most of the CMIP5 models. Among the contributing G4cdnc models, GISS-E2-R and IPSL-CM5A-LR have particularly few low-level clouds in the region around 30°S, see Fig. 1. This is stressed in Schmidt et al. (2014), who compared cloud data from GISS-E2-R to satellite measurements, and found underestimated cloud covers over mid-latitude ocean regions and a particular deficiency in subtropical low clouds. Likewise, Konsta et al. (2016) compared tropical clouds in IPSL-CM5A-LR to satellite observations and found an underestimation of total cloud cover (underestimated low- and mid-level tropical clouds and overestimated high clouds) associated to a high bias in cloud optical depth. Other models have similar issues. For instance, Stevens et al. (2013) showed that MPI-ESM-LR has prominent negative biases in the major tropical stratocumulus regions. Although the low clouds in this study are an approximation as explained in Section 2.2, a comparison between our Fig. 2 and the satellite-based observations of low clouds in Fig. 1 of Cesana and Waliser (2016) clearly demonstrates this. Although MPI-ESM-LR has a globally averaged cloud fraction close to the multi-model average (Table 2), there is a concentration of clouds around the poles rather than at lower latitudes (see Fig. 2). Brightening polar clouds may be less efficient than if the majority of clouds were located at lower latitudes, due to the low solar angle at high latitudes, although influences of cloud changes on the long wave spectrum may still be large (Kravitz et al., 2014). Yet, as we will show in the next section, the climate response of MPI-ESM-LR is still the highest of all the models.

In an evaluation of 19 CMIP5 models against NASA's "A-Train" satellites, Jiang et al. (2012) found a best estimate global mean observed liquid water path (LWP) of 30-50 gm$^{-2}$, with an uncertainty range from 10 to 100 gm$^{-2}$. Table 2 shows a vast inter-model spread in annual average LWP. Values range from 61.7 gm$^{-2}$ (MPI-ESM-R) to 194.7 gm$^{-2}$ (GISS-E2-R). These variations in cloud thickness can be decisive to a model's response to cloud seeding since (given similar levels of cloud condensation nuclei) clouds with LWP above a certain level will be less susceptible to changes in the number of cloud droplets (Sorooshian et al., 2009).

## 4    Climate response to G4cdnc

The G4cdnc experiment results in a model median ERF (calculated using the Gregory regression method, as explained in Section 2.2) of -1.91 [-0.58 to -2.48] Wm$^{-2}$, where the numbers in brackets indicate model minimum and maximum values, see Fig. 3a and Table 3. There is a factor 4.3 difference between the highest and lowest model ERFs; while CSIRO-Mk3L-1-2 has the largest radiative forcing of -2.48 Wm$^{-2}$, GISS-E2-R has the weakest (-0.58 Wm$^{-2}$). The models with a weak ERF typically also have a weak correlation between temperature change and change in TOA radiative flux imbalance (Table S1). The model median geographical pattern of this radiative perturbation is shown in Fig. 4a.

The negative forcing of increasing CDNC cools the near surface air temperatures with a model median of 0.96 [-0.17 to -1.21] K, compared to the RCP4.5 scenario. Figure 3b shows the time series from year 2020 to 2090 of the G4cdnc-RCP4.5 difference in global mean near-surface air temperature. The figure shows that NorESM1-M, GISS-E2-R and IPSL-CM5A-

LR are the models that yield the weakest temperature response and these are the models with the weakest effective radiative forcing. Conversely, MPI-ESM-LR, with the next largest forcing, shows the strongest cooling.

Shown in Table 3 is also the "MCB sensitivity", defined here as the global temperature change normalized by the ERF. We find a model median value of 0.47, with a much smaller inter-model spread (a factor 1.8 difference between highest and lowest model value). MPI-ESM-LR and BNU-ESM give the strongest cooling per degree forcing.

Fig. 4 b shows the geographical pattern of the ensemble median temperature difference between the G4cdnc and *RCP4.5* experiments, based on years 2020-2069 (for individual models, see Fig. S1). There is a strong polar amplification of the cooling signal, with largest cooling over the Arctic from positive sea-ice feedback, and a somewhat weaker cooling around Antarctica. Individual model numbers of the Arctic amplification is given in Table S2, but the median value is 1.9. Some of the models (NorESM1-M, BNU-ESM and MIROC-ESM) show a particularly large spatial correlation (around -0.5 and significant at the 99% level) between the magnitude of the cooling (averaged over 2020-2069) and the baseline (averaged over the 20 first years of the RCP4.5 simulation) low cloud fractions, see Table S1. Such a tendency can also be seen in the ensemble median temperature change of Fig. 4b; typical stratocumulus regions such as parts of the tropical Atlantic ocean and the Pacific ocean off the coasts of Peru and USA (Wood, 2012) show stronger cooling.

Over oceans, the cooling also has a slight tendency to be stronger in regions which have a low baseline LWP. Correlations between the average change in temperature and the baseline LWP for each model gives the individual model correlation coefficients in Table S1, and correlations between grid cell model medians of these quantities gives a  spatial correlation coefficient of 0.42 (significant at the 99% level). Such a tendency might indicate that cloud susceptibility, or the potential of a cloud to produce cooling by increased albedo in response to the increase in droplet numbers, is larger in clouds that are not too dense to begin with. However, two of the models (NorESM1-M and BNU-ESM) have correlations of -0.30 and -0.47, respectively, indicating larger cooling in areas of larger water paths. Note also that changes in LWP are highly variable between models (Table 3 and Fig. S2), with particularly strong positive changes for the two models with prescribed CDNC (MPI-ESM-LR and CSIRO-Mk3L-1-2). Although the CDNC is increased solely over the oceans, the land masses, particularly at low latitudes, cool more than the ocean regions. We find that land areas between 35 °S and 35 °N cool by 1.08 K while low-latitude ocean areas cool by 0.83 K.

As a consequence of the cooling climate, there is a weakening of the hydrological cycle resulting in a decreasing global mean precipitation of -2.35 [-0.57 to -2.96] %. As expected, the model with the strongest cooling has the largest reduction in global precipitation. The total cloud cover increases in all models but BNU-ESM, and there is a strong and statistically significant correlation (coefficient of -0.71) between how much the global mean cloud cover changes for a model and its baseline fraction of low clouds. The model median cloud covers (Fig. 4c) are reduced in high northern and southern latitudes due to the particularly strong cooling and increase in sea-ice in these regions (see Fig. S3 for individual total cloud change patterns). Clouds are also reduced over large regions of mid-latitude land masses, such as over Russia, Northern Europe and North America, and in the inter-tropical convergence zone (ITCZ). Changes in precipitation (Fig. 4d) is mostly correlated to the cloud changes, with notable exceptions being the northern Pacific and Atlantic, where precipitation is reduced in spite of an increase in clouds. In contrast to the marine stratocumulus geoengineering experiment in Jones et al. (2009) only two models (MPI-ESM-LR and IPSL-CM5A-LR) show drying over the Amazon (see individual precipitation change patterns in Fig. S4). Both total cloud cover and precipitation show distinct differences in land/sea responses. Specifically, the cloud cover between 35 °S and 35 °N tends to increase more over land (2.15 %) than over the ocean

(0.32 %). While precipitation increases by 1.2 % over land, there is for the same latitudes a drying of 3.8 % for the oceans, over which the applied geoengineering causes suppression of evaporation (not shown).

Figure 5a shows changes in outgoing short-wave radiation at TOA while Fig. 5b shows outgoing long-wave radiation (OLR). The increase in outgoing short-wave radiation is strongest in ocean regions typically associated with low clouds, and changes little over land. In contrast, the OLR has a band of increases around the ITCZ, but otherwise decreases. The decrease is most pronounced near the poles, but also over tropical land masses. Figure 5c shows that the low cloud cover (see Fig. S5 for individual model changes) generally has a pattern similar to the total cloud cover change, and seems to be a dominant cause of the changes we see in outgoing short-wave radiation (a significant spatial correlation of 0.35 is found between the Figures 5a and 5c). Figure 5d, on the other hand, shows that high clouds increase primarily over tropical and subtropical land masses, producing the reduction in OLR. This is from increasing convection over land, producing more anvils and ice clouds, as also reflected in the enhancement of precipitation in these areas. The model-median spatial correlation between OLR and change in high clouds is -0.55 and highly significant.

The ability of ecosystems to adapt, and the risk for extinction, dramatically increases under rapid climate change (e.g., Jump and Peñuelas, 2005; Menéndez et al., 2006). One of the concerns regarding climate engineering is the possibility that, due to unforeseen events or government issues, there may be a sudden suspension of the climate engineering efforts, casting Earth's climate into a phase of rapid re-warming. This termination effect has been investigated in several studies (Aswathy et al., 2015; Jones et al., 2013; Matthews and Caldeira, 2007). As mentioned in Section 2.1, six models in the present study simulated the termination effect by turning off the CDNC perturbation and continuing the simulations for 20 years. The effect on global temperatures (relative to the RCP4.5 scenario) in this termination period can be seen in the last 20 years in Fig. 3b. At the end of these 20 years, temperatures are almost back at the RCP4.5 levels. Previous GeoMIP publications investigating stratospheric aerosol injections (see, e.g. Berdahl et al., 2014) have noted a much faster warming in the rebound or termination period than in the RCP4.5 scenario, and this we find also here: for years 2070 to 2090 there is a warming of 0.007 K/year for RCP4.5 and a warming of 0.040 K/year for G4cdnc. In a GeoMIP study of stratospheric aerosol injections using three global climate models, Aswathy et al. (2015) find that comparing the mean temperatures of the years 2050-2069 to 2020-2079 (where the latter is the termination period), there was a strong Arctic amplification when looking at the RCP4.5 scenario, but a weaker amplification in the climate engineering scenario. Consistently with this, we find that the model median Artic amplification is 3.6 for the RCP4.5 scenario and 1.7 for G4cdnc, for the same years (see Table S2). The pattern of change in the termination period (Fig. S5) is broadly just a reversal of the geographical patterns seen in Fig. 4. However, the spread between the models, as indicated by the hatching in the maps, is much larger for the termination effect precipitation response than for the temperature response, as found also in Jones et al. (2013).

## 5    Discussion

Increasing CDNC in low marine clouds results in global cooling. This is accompanied by a global mean reduction in precipitation. These signals are robust across all nine models, but the magnitude of the change varies by 50 % between models for the temperature change, and by 40 % for the precipitation change.

Although the CDNC is only increased over oceans, we find stronger cooling over land masses, particularly in the lower latitudes. High clouds increase the most over low-latitude land, and the OLR is reduced over low-latitude land and increases

over oceans. This is consistent with a shift in convection from ocean to land, which also explains the increase in precipitation of +1.2 % over low-latitude land as opposed to a drying of -3.8 % over oceans. This kind of pattern has been seen in previous modelling studies of marine cloud brightening. For instance, Bala et al. (2011) used an Earth system model to perform simulations where the effective droplet size in liquid clouds was reduced over oceans globally. They found that the differential enhancement of albedo over oceans and land triggered a monsoonal circulation with rising motion over land and sinking motion over oceans. This happened as the vertically integrated air mass cooled more over the oceans where the cloud albedo was increased than over land. The resulting density change caused an increase in net land to ocean transport above 400 hPa and an increase in the net ocean to land transport below 400 hPa. Alterskjær et al. (2013) analyzed an ensemble of three Earth system models that are all included in the present study and noted the same land-sea difference in warming and associated hydrological cycle changes. Similarly, Niemeier et al. (2013) compare three types of solar radiation management (stratospheric sulfur injections, mirrors in space and MCB), and found that only the MCB experiment induced changes to the Walker circulation.

Marine cloud brightening through emissions of some agent, for instance sea salt, into the marine boundary layer influences the low-lying clouds. But we find that the amount and location of clouds vary greatly between the models, which will cause variation in their response to MCB (see e.g. Chen and Penner (2005), who found cloud fraction to be one of the most important sources of uncertainty in model-differences in estimates of the first indirect effect). We also find a substantial inter-model spread in the LWP, which is one of the factors determining how susceptible a cloud is to albedo changes through the Twomey effect (Twomey, 1974). For instance, MPI-ESM-LR has a fraction of low clouds that is close to the model average, but a geographical distribution of those low clouds that could imply a reduced efficiency of the CDNC enhancement. Low clouds at higher latitudes are less effective at cooling, as the solar angle and hence the climate effect, is lower. MPI-ESM-LR also has the lowest globally averaged LWP of all models. The potential that an increase in CDNC has to enhance the cloud albedo (the cloud albedo susceptibility) has been shown to be smaller in clouds with low LWP. For instance, Painemal and Minnis (2012) investigated satellite data for typical stratocumulus regions and found an increase in cloud albedo susceptibility with LWP up to about 60 $gm^{-2}$, after which the susceptibility levelled off or even decreased slightly. Even so, this model has the second strongest ERF of all the models, possibly due to a very strong increase in cloud amounts between around 40 °S to 40 °N. GISS-E2-R, on the other hand, is the model with the weakest ERF. We find that this model has the smallest low cloud fractions among all the models, and there is also a tendency for these clouds to be concentrated at higher latitudes. In addition, the LWP is extremely high, which could conceivably mean that the clouds are already relatively saturated with respect to further changes in droplet numbers, in line with the findings of Painemal and Minnis (2012) mentioned above.

Geoffroy et al. (2017) investigated the role of different stratiform cloud schemes to the inter-model spread of cloud feedbacks, looking at 14 models – four of which were used in the present study. They found that NorESM1-M, which diagnoses stratiform clouds based on relative humidity and atmospheric stability, had an opposite cloud feedback from models (including HadGEM2-ES, CSIRO-Mk3L-1-2 and IPSL-CM5A-LR) that diagnoses such clouds based solely on relative humidity. BNU-ESM, whose change in TOA radiative flux imbalance is that of the model median, uses the same stratiform cloud scheme as NorESM1-M (Ji et al., 2014), and like NorESM1-M the change in low cloud cover is near zero. It does not, however, have the negative change in LWP seen in NorESM1-M. This may be due to the specific set-up of the simulations in BNU-ESM, where to obtain the 50% decrease in CDNC a direct alteration of liquid droplet size was done. In terms of TOA radiative flux imbalance, CSIRO-Mk3L-1-2 has the strongest response of all models. LWP, amount of

low clouds, and the geographical distribution of low clouds, however, are all similar to the model average. The model has the strongest cooling per forcing (ERF), and shows the strongest increase in low cloud cover.

As seen in Table 1, there are large differences between models as to how the aerosol-cloud processes are parametrized, and this presumably has an impact on the MCB climate response. For instance, Morrison et al. (2009) found that the complexity of the cloud schemes had large impacts on stratiform precipitation; Penner et al. (2006) compared several models and concluded that the method of parameterization of CDNC can have a large impact on the calculation of the first indirect effect. CSIRO-Mk3L-1-2 and MPI-ESM-LR are the only models that have prescribed CDNC levels in their simulations, and these models stand out as having ERFs well above any of the other models. While an evaluation of which liquid cloud parametrizations are more appropriate is beyond the scope of this paper, this might be an indication that prescribing CDNC levels may lead to exaggerated responses to marine cloud brightening.

A caveat of the present study is that the spatial resolution of global climate models is too coarse to resolve important processes such as convection or precipitation formation. These processes are instead parametrized, which may lead to unknown artifacts in the responses, dependent on the specific formulations used (see e.g. Clark et al. (2009) who compared precipitation forecast skills for convection-allowing and convective-parametrized ensembles). It is also important to point out that the G4cdnc experiment was not designed to give a realistic representation of the magnitude of cooling and other climate responses to MCB, but rather to test the robustness of models in simulating geographically heterogeneous radiative flux changes and to see their effects on climate. Increasing CDNC by 50% over all oceans is clearly an exaggeration of what could credibly be done. A more realistic GeoMIP marine cloud brightening experiment, G4sea-salt, is analyzed by Ahlm et al. (2017). Their results show what areas in the participating models where the sea salt seeding is the most effective at brightening the clouds. This includes the decks of persistent low level clouds off the west coasts of the major continents in the sub-tropics. Alterskjær et al. (2012) investigated the susceptibility of marine clouds to MCB based on satellite as well as model data, and reached a similar conclusion; large regions between 30°S and 30°N, and especially clean regions in the Pacific and Indian oceans, along with regions in the western Atlantic, were susceptible.

## 6    Summary and conclusion

Nine models have conducted a coordinated idealized marine cloud brightening experiment G4cdnc of the GeoMIP project, producing a median ERF of -1.91 [-0.58 to -2.48] Wm$^{-2}$. While climate in general cools as intended, there are large geographical variations in the climate response. For instance, although the global cooling leads to a general drying, robust model responses also include a slight precipitation increase over low-latitude land regions, such as subtropical Africa, Australia and large parts of South America.  This is a result of the land-sea contrast in temperature initiated by brightening clouds only over ocean regions, which results in a circulation pattern with rising motion over land and sinking air over ocean, particularly over lower latitudes.

Responses in precipitation and clouds show particularly large inter-model spreads. For some models we find a strong dependency of the climate response on the location of low clouds in the baseline simulation, RCP4.5, and also on the thickness in terms of LWP of the clouds. For other models no such clear dependency is found. Variations in the complexity of microphysical schemes contribute to the large model spread in responses, with an apparent tendency for stronger responses for the more simple representations. Conceivably, the more realistically the processes between CDNC

perturbation and cloud albedo change are simulated, the more buffering processes are included, dampening the total climate response (Stevens and Feingold, 2009). Indeed, climate changes from G4cdnc do not include all processes in play between seeding of the clouds and the actual climate response even in the models with the most advanced microphysics schemes. While the increase in CDNC is assumed to originate from an increase in for instance sea salt particles in the marine boundary layer, one caveat with the G4cdnc experiment is that it does not simulate these particles specifically. This is however done in the follow-up experiment G4sea-salt, which was performed by three of the models investigated here. Consistent with previous findings (Jones and Haywood, 2012; Partanen et al., 2012), Ahlm et al. (2017) found for G4sea-salt that the direct effect (aerosol-radiation effect) of the sea salt aerosols themselves contribute significantly to the total radiative effect of sea-salt climate engineering. It should be noted that even experiments that capture this direct effect may be subject to biases caused by processes not included in the experiments. For instance, Stuart et al. (2013) noted that sea-spray climate engineering studies assume a uniform distribution of the emitted sea-salt in ocean grid boxes, which does not account for sub-grid aerosol coagulation within sea-spray plumes. They find that accounting for this effect reduces the CDNC (and the resulting radiative effect) by about 50 % over emission regions, with variations from 10 to 90 % depending on meteorological conditions.

Cloud feedbacks remain the largest source of inter-model spread in predictions of future climate change (Vial et al., 2013). While detailed, high-resolution simulations of marine sky brightening give crucial insights into processes that contribute to the total climate response, multi-model idealized experiments such as the G4cdnc are still important to untangle the cloud responses. We hypothesize that liquid cloud parameterizations ought to be of appropriate complexity in order to attempt to model marine cloud brightening and the climate response.

**Acknowledgements**

The work of JEK, CWS, HM and LA was supported by the EXPECT project, funded by the Norwegian Research Council, grant no. 229760/E10. CWS wishes to thank CICERO for letting her continue and complete the present project. HM was also funded by RCN grant 261862/E10. LA was also supported by the Swedish Research Council FORMAS (grant 2015-748). The Pacific Northwest National Laboratory is operated for the U.S. Department of Energy by Battelle Memorial Institute under contract DE-AC05-76RL01830. SJP was supported by the Australian Research Council's Special Research Initiative for the Antarctic Gateway Partnership (Project ID SR140300001). The work of DJ and JCM was supported by the National Basic Research Program of China grant number 2015CB953600.

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

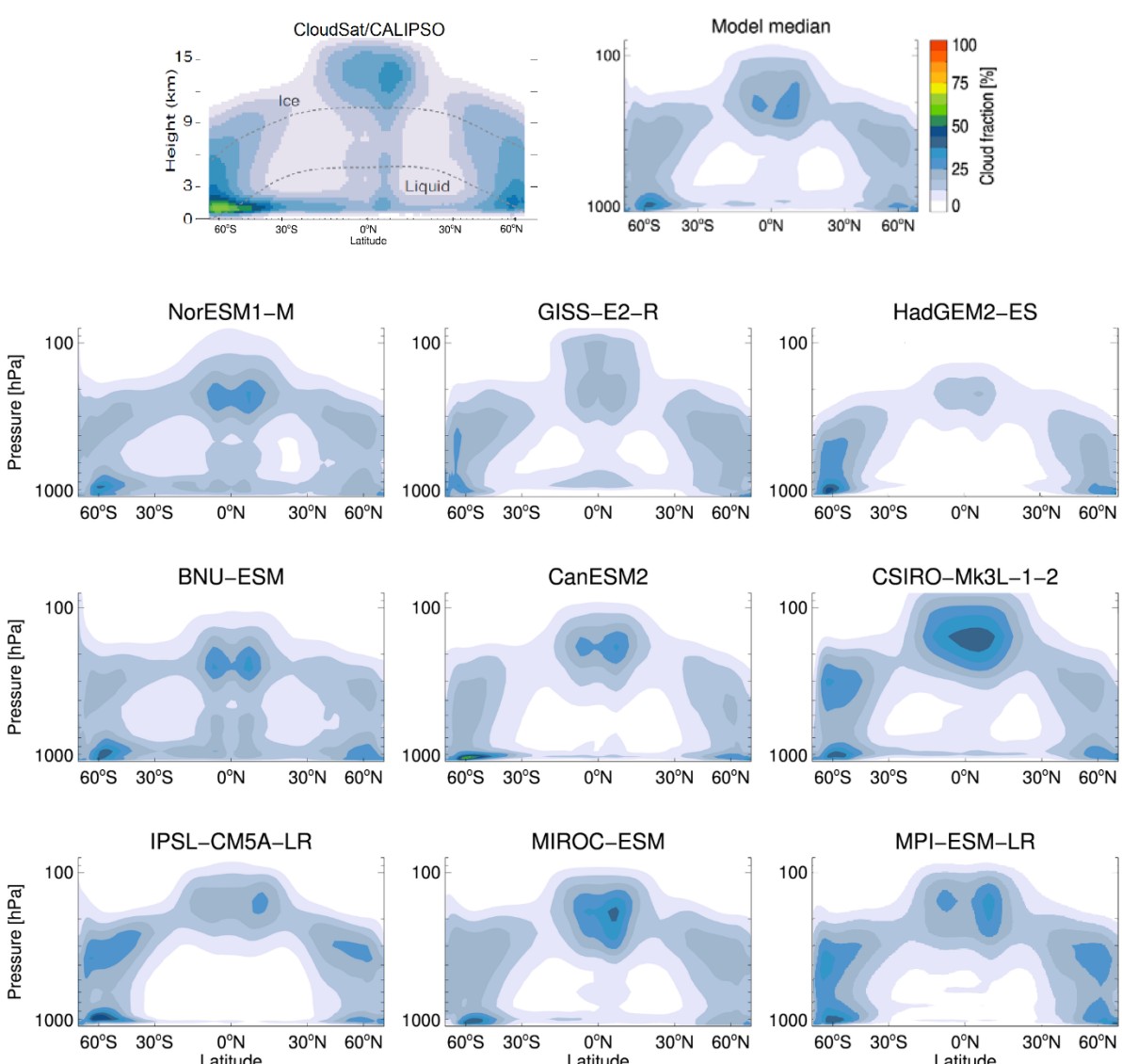

**Figure 1: Upper left panel shows cloud fraction [%] from CloudSat/CALIPSO (upper left panel), adapted with permission from Fig. 7.5 (c) from Boucher et al. (2013). Upper right panel shows the GeoMIP model median for the first 20 years of the RCP4.5 scenario (upper right panel), and remaining panels show individual RCP4.5 GeoMIP model results for the same years. Note different vertical axes for the AR5 and GeoMIP plots.**

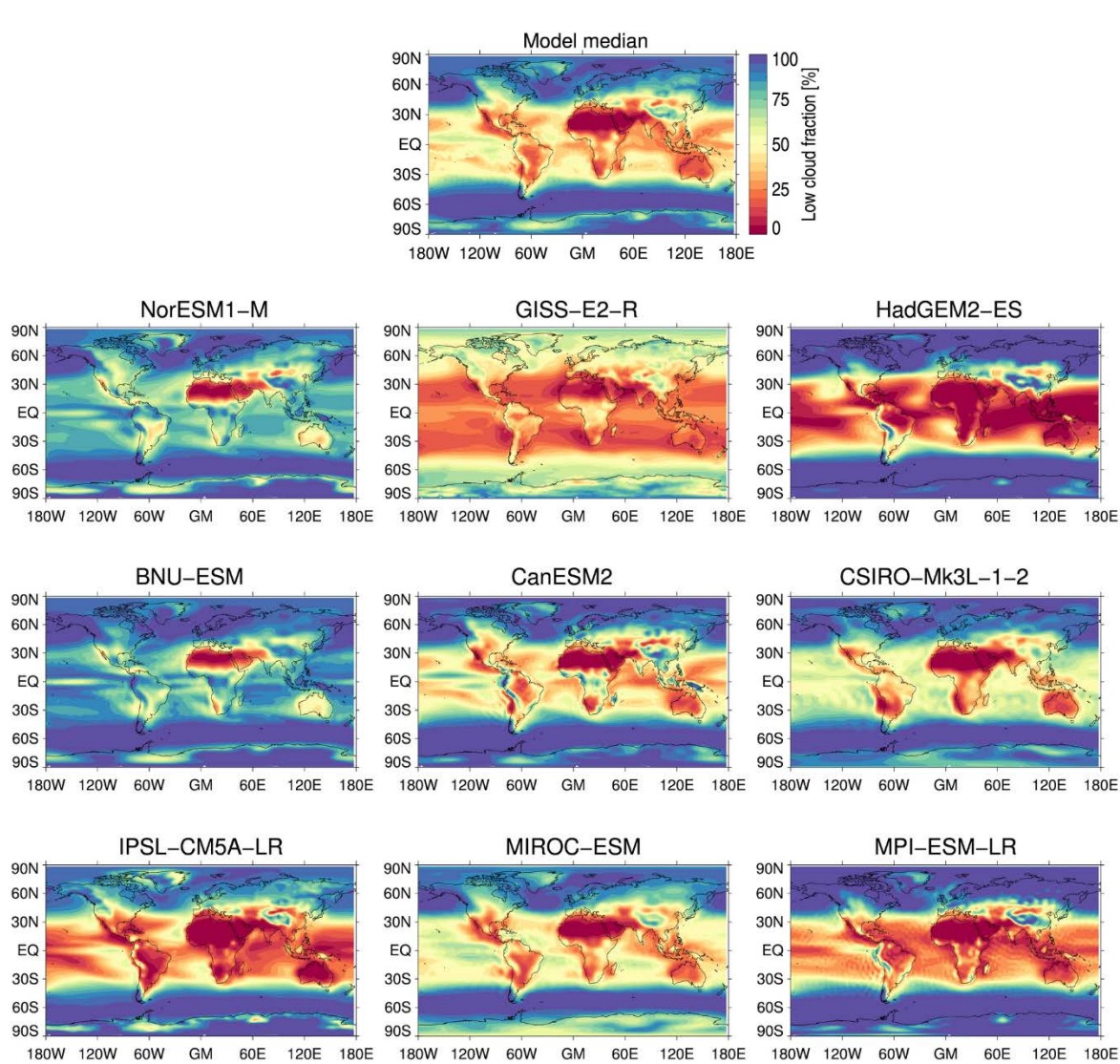

**Figure 2: Low cloud fraction [%] for the first 20 years of the RCP4.5 scenario, for the model median as well as the individual models. Low clouds are estimated using a random overlap assumption (Tian and Curry, 1989).**

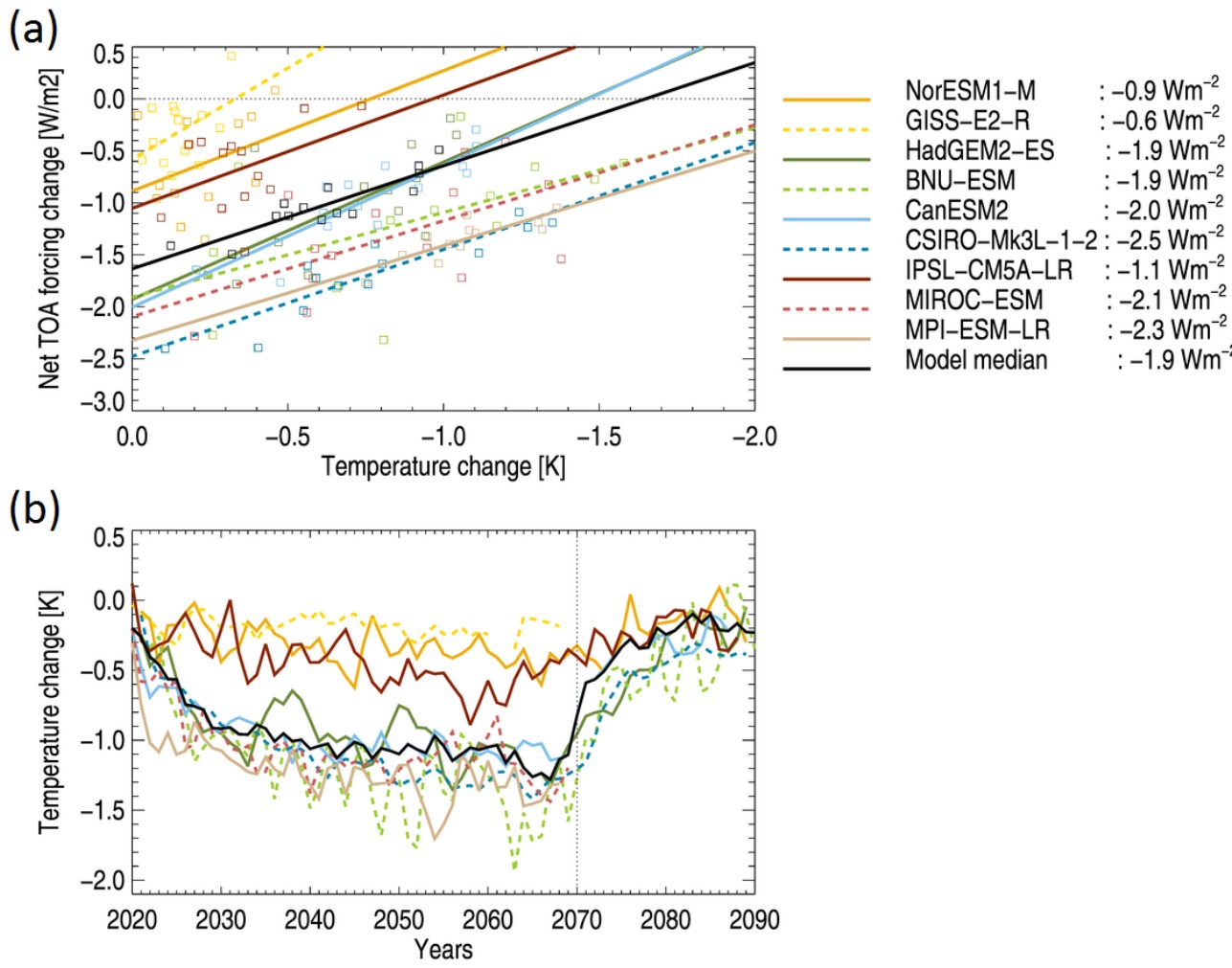

**Figure 3: a) Regression of global annual means of the net TOA radiative flux imbalance and near surface temperatures for each model cf. Gregory et al. (2004). Each square represent global annual mean for each of the first ten years and decadal means for the remaining part of the simulations (i.e. last four decades, 2030 - 2069). Numbers to the right gives the intercept – i.e., the effective radiative forcing. b) Time series of the difference in global annual mean near-surface temperature between G4cdnc and *RCP4.5* for each model. Dotted vertical line indicates the onset of the termination period.**

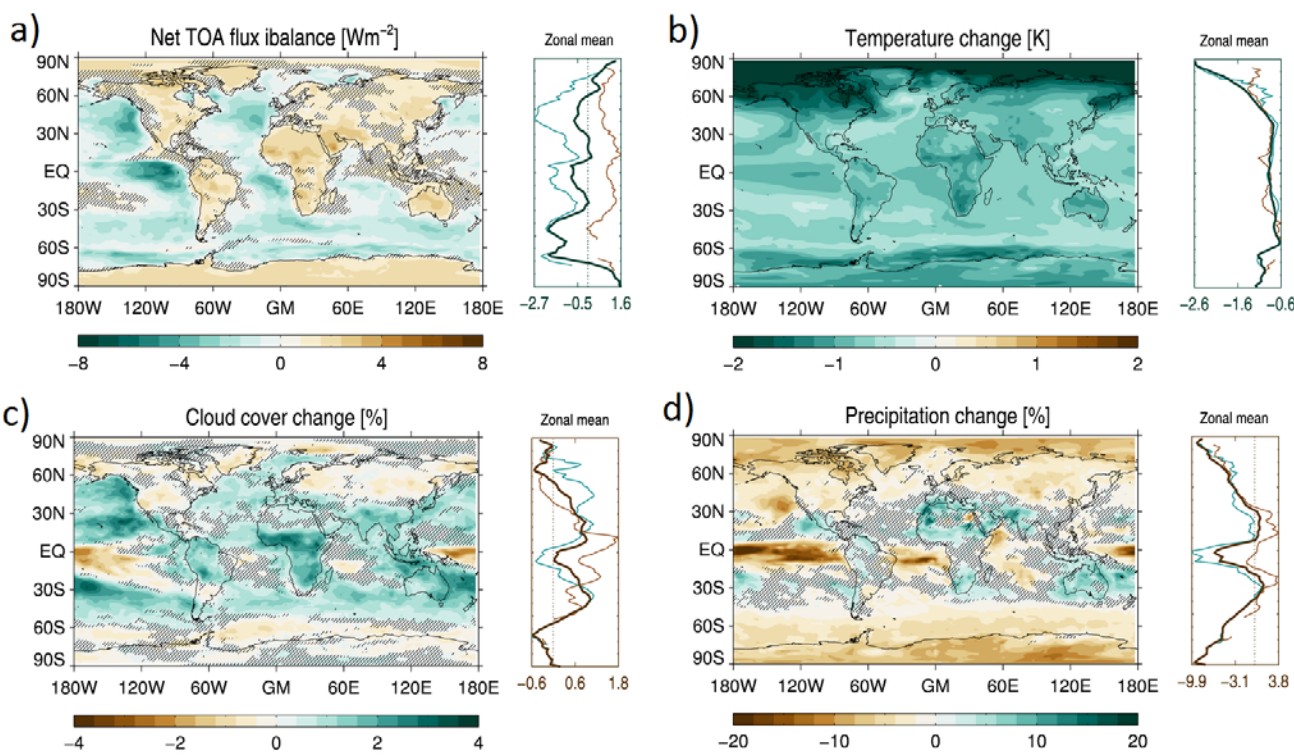

**Figure 4: Ensemble median (taken in each grid cell) G4cdnc-RCP4.5 difference based on years 2020-2069 of a) TOA net radiative flux imbalance (Wm$^{-2}$), b) near-surface air temperature change (K), c) total cloud cover (%), and d) precipitation (%). Hatched areas are grid cells where less than 75 % of the models agreed on the sign of the change. Zonal averages are given to the right of each panel, where brown and blue lines indicate land-only and ocean-only averages. See figures S-2 and S-3 for individual model changes in cloud cover and precipitation, respectively.**

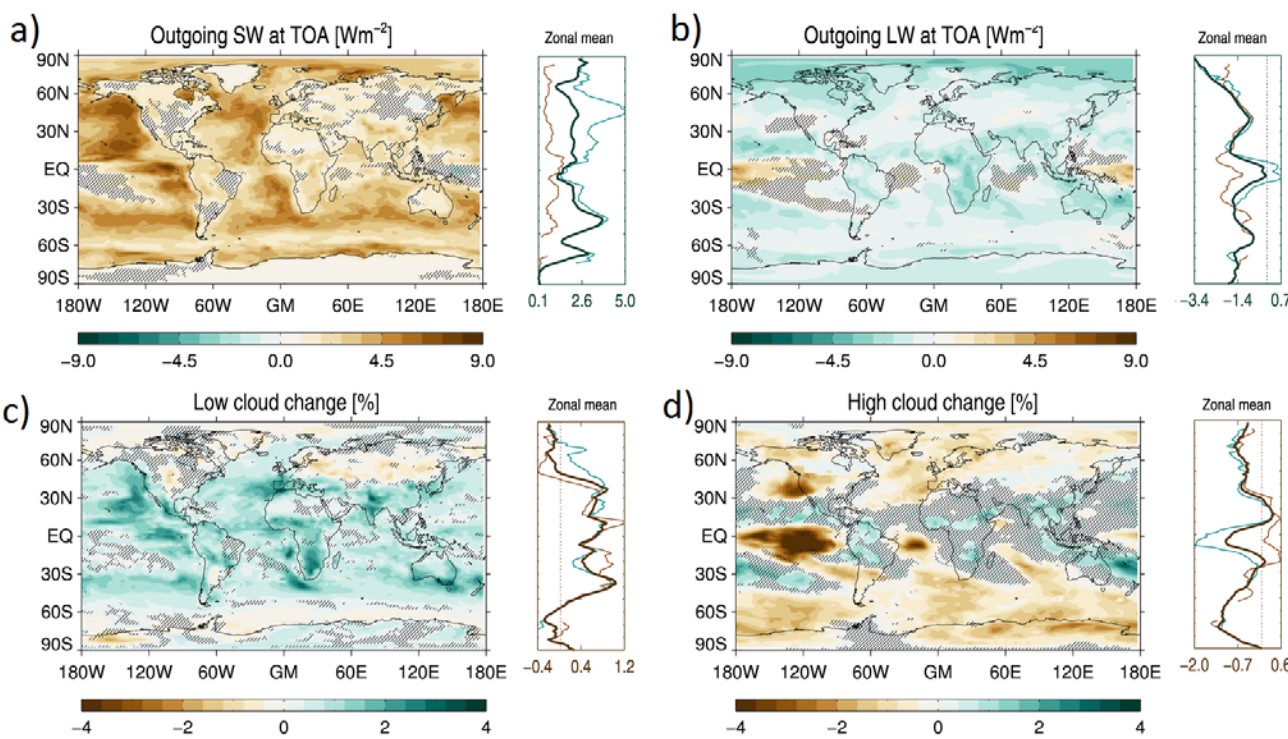

**Figure 5: Ensemble median (taken in each grid cell) G4cdnc-RCP4.5 difference of years 2020-2069 of a) outgoing short-wave radiation at TOA (Wm⁻²), b) outgoing long-wave radiation at TOA (Wm⁻²), c) change in low cloud fraction (%), and d) change in high cloud fraction (%). Hatched areas are grid cells where less than 75 % of the models agreed on the sign of the change. Zonal averages are given to the right of each panel, where brown and blue lines indicate land-only and ocean-only averages, respectively.**

| Model | No. of gridcells (lat x lon) | No. of vert layers (type) | RCP4.5 / G4cdnc realizations | Representation of aerosol indirect effect |
|---|---|---|---|---|
| BNU-ESM | 64 x 128 | 26 (hybrid sigma) | 1 / 1 | Single-moment microphysics scheme; Rasch and Kristjánsson (1998) with modification by Zhang et al. (2003). NOTE: To achieve effects of 50% increase of CDNC over ocean regions below 680 hPa, a direct alteration of liquid droplet size by dividing (1.5^(1/3)) is done. |
| CanESM2 | 64 x 128 | 35 (hybrid sigma) | 5 / 3 | Prognostic microphysics scheme accounting for the first indirect effect but not the second indirect effect (von Salzen et al., 2013) |
| CSIRO-Mk3L-1-2 | 56 x 64 | 18 (hybrid sigma) | 3 / 3 | Prescribed CDNC |
| GISS-E2-R | 90 x 144 | 21 (hybrid sigma) | 3 / 3 | Prognostic calculations of CDNC (Menon et al., 2010), based on Morrison and Gettelman (2008) |
| HadGEM2-ES | 145 x 192 | 38 (hybrid height) | 4 / 1 | Diagnostic CDNC scheme based on Jones et al. (2001) |
| IPSL-CM5A-LR | 96 x 96 | 39 (hybrid sigma) | 4 / 1 | CDNC is computed from the total mass of soluble aerosol through the prognostic equation from Boucher and Lohmann (1995) |
| MIROC-ESM | 64 x 128 | 80 (hybrid sigma) | 1 / 1 | Prognostic calculation of CDNC (Abdul-Razzak and Ghan, 2000) |
| MPI-ESM1-LR | 96 x 192 | 47 (hybrid sigma) | 1 / 1 | Prescribed CDNC |
| NorESM1-M | 96 x 144 | 26 (hybrid sigma) | 1 / 1 | Double-moment microphysics scheme with prognostic calculation of CDNC (Abdul-Razzak and Ghan, 2000; Hoose et al., 2009; Morrison and Gettelman, 2008) |

**Table 1: Information on the contributing models, including resolution, number of realizations and their representation of the aerosol indirect effect.**

| | Total cloud cover [%] | Precipitation [mm/day] | Liquid water path [gm⁻²] | Low cloud cover [%] |
|---|---|---|---|---|
| BNU-ESM | 52.74 | 2.89 | 151.15 | 77.54 |
| CanESM2 | 60.80 | 2.78 | 119.07 | 57.69 |
| CSIRO-Mk3L-1-2 | 66.98 | 2.76 | 118.38 | 59.97 |
| GISS-E2-R | 61.01 | 3.22 | 192.71 | 31.19 |
| HadGEM2-ES | 53.32 | 3.09 | 92.14 | --- |
| IPSL-CM5A-LR | 56.67 | 2.76 | 68.11 | 46.06 |
| MIROC-ESM | 50.37 | 2.82 | 139.28 | 55.88 |
| MPI-ESM-LR | 61.93 | 2.88 | 62.85 | 49.59 |
| NorESM1-M | 53.94 | 2.83 | 142.47 | 73.29 |
| **Model average and spread** | 57.5 (± 5.4) | 2.9 (± 0.2) | 120.7 (± 41.6) | 55.9 (± 23.4) |
| **Observations** | 68 (± 3) [a] | 2.68 [b] | 30-50 [10 to 100] [c] | (not comparable to observations) |

**Table 2: Global averages for the 20 first years (2006-2025) of the RCP4.5 experiment. Model spread is given as one standard deviation. "Low cloud cover" is calculated using a random overlap assumption for all vertical levels below 680 hPa. [a] As averaged from daytime measurements over several remote sensing datasets, see Stubenrauch et al. (2013). [b] Climatology for 1951-2000 from the Global Precipitation Climatology Centre (Schneider et al., 2014). [c] Observations from NASA's "A-Train" satellite observations (Jiang et al., 2012)**

| | Gregory regression ERF | MCB sensitivity | Total cloud cover change | Temp. change | Precip. change | Liquid water path change | Low cloud cover change |
|---|---|---|---|---|---|---|---|
| *Units* | *Wm⁻²* | *KWm⁻²* | *%* | *K* | *%* | *gm⁻²* | *%* |
| BNU-ESM | -1.91 | 0.61 | -0.46 | -1.16 | -2.72 | +1.83 | -0.01 |
| CanESM2 | -2.00 | 0.48 | +0.29 | -0.96 | -2.28 | -0.71 | +1.82 |
| CSIRO-Mk3L-1-2 | -2.48 | 0.43 | +1.17 | -1.07 | -2.93 | +4.52 | +2.00 |
| GISS-E2-R | -0.58 | 0.29 | +0.05* | -0.17 | -0.61 | -0.93 | +0.16 |
| HadGEM2-ES | -1.93 | 0.49 | +1.01 | -0.96 | -2.34 | +0.01 | --- |
| IPSL-CM5A-LR | -1.05 | 0.42 | +1.03 | -0.44 | -1.70 | -1.10 | +1.28 |
| MIROC-ESM | -2.10 | 0.50 | +0.36 | -1.06 | -2.51 | -4.62 | +1.55 |
| MPI-ESM-LR | -2.32 | 0.52 | +0.78 | -1.21 | -2.96 | +6.98 | +5.95 |
| NorESM1-M | -0.89 | 0.35 | +0.01 | -0.31 | -0.57 | -0.61 | +0.16 |
| **Ensemble median** | -1.91 (± 0.63) | 0.47 (± 0.09) | +0.18 (± 0.42) | -0.96 (± 0.45) | -2.35 (± 0.92) | -0.61 (± 3.43) | +1.55 (± 1.93) |

**Table 3: G4cdnc minus RCP4.5 difference (based on years 2020-2069) in the key variables, including the effective radiative forcing as estimated in Figure 3 a). An asterisk denotes that the change is not significant at the 95 % level by the Kolmogorov-Smirnov test.**