# Peer review of "Response to marine cloud brightening in a multi-model ensemble"

_Atmospheric Chemistry and Physics, 2017_

## Referee Comment (RC1) · Anonymous Referee #1 · 25 Aug 2017

**Reviewer Comments:**

**Major Revisions:**
The authors present the results from Earth System Model simulations from the marine cloud brightening experiment of the Geoengineering Model inter comparison project. Marine cloud brightening (MCB), one of several solar radiation management (SRM) to mitigate the climate change. MCB via 50% increase in the CDNC of low clouds over the global oceans shows an effective radiative forcing of $-1.9Wm^{-2}$, the global temperature decrease of 0.95K, and a significant decrease in global precipitation. Although authors show that the MCB mitigates the surface temperature, the termination effect is not discussed in detail. As far as climate engineering is concerned, termination effect

has large significance. Also, it would be interesting to see the effect of MCB on climate extremes.

The topic- Response to marine cloud brightening in a multi-model ensemble- is of interest and it fits the scope of ACP. The paper can be accepted with major revision. Please see the major comments below.

**Major comments:**

• Out of nine models, six models investigated the termination effect, so it would be interesting to see the terminating effect in those models. A similar study which uses sea salt aerosols for MCB reported that termination effect results in an increase in precipitation and extremes (Aswathy et al., 2015). So termination effect, especially on precipitation, cloud cover, and temperature (spatial and zonal averages) could be included in the revised manuscript.
• Also it would be interesting to see the termination effect on polar amplification. A separate section can be included.

**Minor comments:**
• Page 4. l8: Modeled clod climatologies, is it Section 2.3.?
• Figure 1. Indicate a reference mark/line for low clouds (at 680 mb).
• Figure 2. Please consider, including observation.

---

## Referee Comment (RC3) · Anonymous Referee #2 · 5 Sep 2017

The paper is a multi-model comparison to understand marine cloud brightening, including regional responses in cloud and precipitation fields, based on increasing the CDNC droplet number concentration over low clouds by 50% and the RCP4.5 emissions scenario. A radiative forcing (ERF) of -1.9 W m$^{-2}$ is found in the ensemble, with the spread equal to -0.6 to -2.5 W m$^{-2}$.

Generally I thought the style of writing, and the quality of the figures is fine and would support publication in a modified form. The main issues I have with the paper in its current form are (1) it is unclear what has been done in parts of the manuscript; (2) as presented the results may be mis-interpreted unless more care / accuracy is taken in the wording.

The G4CDNC experiment is used. I am not familiar with this experiment myself, and I

had to do some searching in the text to find that it is a 50% increase in CDNC of low clouds. This should be brought to the fore, so that it is clear. It would be useful to know how much of an overestimate applying the CDNC increase to all low clouds over the ocean may be to those that could realistically be subjected to a geoengineering scheme. What area of the earth does this correspond to in each of the different models? This is an important parameter to know: generating the requisite spray for MCB is no trivial task in terms of energy and scale-of-operation requirements.

In the introduction there is reference to the finding of Alterskjaer et al., that seeding may lead to warming under certain conditions. Connolly et al. (2014, Phil Trans) discuss a similar finding in some detail. In short, the finding is that the Abdul Razzak et al parametrization does not work outside of the region they were originally tested and, in some ranges of parameter space, increasing the aerosol number concentration can lead to a spurious reduction in droplet number concentration. I note from table 1 that some of the models used in this study employ the same parameterisation, including NorESM. While CCN activation schemes are not a focus of this particular paper, some of the reasons for the effects talked about in the introduction may be due to CCN activation schemes, so it is an important point for other people working in this area.

I was not familiar with the Gregory regression method (section 3) until I did a search on the Internet. I think the original source should be cited. Would it be more complete to show / present the correlation coefficients associated with the analysis, to allow the reader to assess its suitability?

Fig 4a was not referred to directly in the text.

Although the models in the paper are important tools for climate assessment, the major issue with this kind of analysis is that the processes under investigation are not resolved at high enough grid resolution. There should be some caveat in the paper so that readers who are unfamiliar with the details are not mis-lead.

The statement: "Our results suggest that liquid cloud parameterizations ought to be of

appropriate complexity in order to attempt to model marine cloud brightening and the climate response." I did not see how this conclusion was arrived at. How do you know that liquid cloud parameterisations are complex enough?

---

## Author Comment (AC1) · 17 Oct 2017

**Response to reviewers**

We thank the reviewers for their comments and efforts towards improving this manuscript. Below, the reviewers' comments are given in blue italic font, and our response follows in black font. Text in small font are excerpts from the new manuscript.

**Reviewer #1**
* * *
*Out of nine models, six models investigated the termination effect, so it would be interesting to see the terminating effect in those models. A similar study which uses sea salt aerosols for MCB reported that termination effect results in an increase in precipitation and extremes (Aswathy et al., 2015). So termination effect, especially on precipitation, cloud cover, and temperature (spatial and zonal averages) could be included in the revised manuscript.*

Thank you for this assessment. We had chosen not to emphasize the termination period because we wanted to keep the focus on the effect of performing MCB. We do, however, agree with the reviewer that the climate response to a sudden MCB termination is an important aspect. We have therefore added a new figure and a table in the Supplement, showing temperatures, precipitation, cloud cover and TOA radiation fluxes. We find that the pattern of the changes in temperature, precipitation and cloud cover during the termination period is to a large extent a reversal of the pattern of change resulting from performing MCB in the first place. As already stated in the text, however, the rate of change is higher in the termination period. We have now added the following paragraph at the end of Section 4:

The ability of ecosystems to adapt, and the risk for extinction, dramatically increases under rapid climate change (e.g., Jump and Peñuelas, 2005; Menéndez et al., 2006). One of the concerns regarding climate engineering is the possibility that, due to unforeseen events or government issues, there may be a sudden suspension of the climate engineering efforts, casting Earth's climate into a phase of rapid rewarming. This termination effect has been investigated in several studies (Aswathy et al., 2015; Jones et al., 2013; Matthews and Caldeira, 2007). As mentioned in Section 2.1, six models in the present study simulated the termination effect by turning off the CDNC perturbation and continuing the simulations for 20 years. The effect on global temperatures (relative to the RCP4.5 scenario) in this termination period can be seen in the last 20 years in Fig. 3b. At the end of these 20 years, temperatures are almost back at the RCP4.5 levels. Previous GeoMIP publications investigating stratospheric aerosol injections (see, e.g. Berdahl et al., 2014) have noted a much faster warming in the rebound or termination period than in the RCP4.5 scenario, and this we find also here: for years 2070 to 2090 there is a warming of 0.07 K/decade for RCP4.5 and a warming of 0.40 K/decade for G4cdnc. In a GeoMIP study of stratospheric aerosol injections using three global climate models, Aswathy et al. (2015) find that comparing the mean temperatures of the years 2050-2069 to the years 2020-2079 (where the latter is the termination period), there was a strong Arctic amplification when looking at the RCP4.5 scenario, but a weaker amplification in the climate engineering scenario. Consistently with this, we find that the model median Arctic amplification is 3.6 for the RCP4.5 scenario and 1.7 for G4cdnc, for the same years (see Table S2). The pattern of change in the termination period (Fig. S6) is broadly a reversal of the geographical patterns seen in Fig. 4. However, the spread between the models, as indicated by the hatching in the maps, is much larger for the termination effect precipitation response than for the temperature response, as found also in Jones et al. (2013).

[Figure]

**Figure S6:** Termination effect. First, the mean change for the G4cdnc years 2050-2069 (the last 20 years of the geoengineering period) is subtracted from the termination period 2070-2089, and the difference for the corresponding years of the RCP4.5 simulations are then subtracted from this number. This gives us an estimate of how much stronger the climate change from abrupt suspension of geoengineering is than the change for the corresponding period for RCP4.5. Panels show ensemble median (taken in each grid cell) differences in a) TOA net radiative flux imbalance (Wm$^{-2}$), b) near-surface air temperature change (K), c) total cloud cover (%), and d) precipitation (%). Hatched areas are grid cells where less than 75 % of the models agreed on the sign of the change. Zonal averages are given to the right of each panel, where brown and blue lines indicate land-only and ocean-only averages.

*Also it would be interesting to see the termination effect on polar amplification. A separate section can be included*

We thank the reviewer for this suggestion. We have now included a Table S2, where the Arctic amplification for the temperature difference of (2070-2089) minus (2050-2069) is given for both RCP4.5 and G4cdnc and added a discussion of this in Section 4 (see sentences below, which is an excerpt from the paragraph shown in the previous review point):

In a GeoMIP study of stratospheric aerosol injections using three global climate models, Aswathy et al. (2015) find that comparing the mean temperatures of the years 2050-2069 to 2070-2089 (where the latter is the termination period), there was a strong Arctic amplification when looking at the RCP4.5 scenario, but a weaker amplification in the climate engineering scenario. Consistently with this, we find that the model median Artic amplification is 3.6 for the RCP4.5 scenario and 1.7 for G4cdnc, for the same years (see Table S2).

|  | MCB period | Termination period, RCP4.5 | Termination period, G4cdnc |
|---|---|---|---|
| BNU-ESM | 6.6 | 6.6 | 1.7 |
| CanESM2 | 3.6 | 3.6 | 2.0 |
| CSIRO-Mk3L-1-2 | 3.8 | 3.8 | 0.6 |
| GISS-E2-R | 2.1 | 2.1 | --- |
| HadGEM2-ES | 3.8 | 3.8 | 0.9 |
| IPSL-CM5A-LR | 1.9 | 1.9 | 2.0 |
| MIROC-ESM | 1.1 | 1.1 | --- |
| MPI-ESM-LR | --- | --- | --- |
| NorESM1-M | 2.1 | 2.1 | -0.5 |
| **Median** | 3.6 | 3.6 | 1.7 |

**Table S2:** The first column give the Arctic amplification for the MCB period 2020-2060, calculated as the difference between G4cdnc and RCP4.5. The next column show for RCP4.5 and G4cdnc, respectively: the Arctic amplification in the termination period, estimated as the mean temperature change (average of 2070-2089 minus average of 2050-2069) for the Arctic (defined as all area above 60° N) divided by the globally averaged temperature change. Leftmost column shows numbers for RCP4.5 while rightmost column shows changes for G4cdnc. Models that did not simulate the termination period is marked with '---'.

*Page 4. l8: Modeled clod climatologies, is it Section 2.3.?*
Thank you for noticing this error – this is actually Section 3! This has now been fixed.

*Figure 1. Indicate a reference mark/line for low clouds (at 680 mb).*
We agree that this would improve the readability of this figure. Lines at 680 hPa are now added as suggested.

[Figure]

[Figure]

[Figure]

**Figure 1: Upper left panel shows cloud fraction [%] from CloudSat/CALIPSO (upper left panel), adapted with permission from Fig. 7.5 (c) from Boucher et al. (2013). Upper right panel shows the GeoMIP model median for the first 20 years of the RCP4.5 scenario (upper right panel), and remaining panels show individual RCP4.5 GeoMIP model results for the same years. Note different vertical axes for the AR5 and GeoMIP plots. Dashed lines mark the 680 hPa level.**

*Figure 2. Please consider, including observation.*

We appreciate the reviewer's suggestion, and we agree that it would be informative to be able to compare these panels to observations, as in Figure 1. However, the low cloud fraction as used in this work is approximated using a random overlap assumption based on monthly mean data. While this provides insight into the geographical distribution of low clouds, as well as how these amounts and distributions vary between models, we do not believe that this quantity will be directly comparable to observations. We have therefore chosen not to provide any observation-based map in Figure 2. However, we have added a reference to Cesana and Waliser (2016, GRL), who show a map of annual mean observationally based low clouds;

Although the low cloud amounts in this study are an approximation as explained in Section 2.2, a comparison between Fig. 2 and the satellite-based observations of low clouds in Fig. 1 of Cesana and Waliser (2016) clearly demonstrates this.

---

## Author Comment (AC2) · 17 Oct 2017

**Response to reviewers**

We thank the reviewers for their comments and efforts towards improving this manuscript. Below, the reviewers' comments are given in blue italic font, and our response follows in black font. Text in small font are excerpts from the new manuscript.

**Reviewer #2**
* * *
*Generally I thought the style of writing, and the quality of the figures is fine and would support publication in a modified form. The main issues I have with the paper in its current form are (1) it is unclear what has been done in parts of the manuscript; (2) as presented the results may be mis-interpreted unless more care / accuracy is taken in the wording*

We thank the reviewer for raising this general concern, and for the opportunity to improve our manuscript by making both methods and interpretation of results more accurate and transparent. We follow the reviewer's suggestions below to make these improvements.

*The G4CDNC experiment is used. I am not familiar with this experiment myself, and had to do some searching in the text to find that it is a 50% increase in CDNC of low clouds. This should be brought to the fore, so that it is clear.*

To clarify at an earlier stage that it is the G4cdnc experiment we focus on in this paper (and also what this experiment involves), we have made changes to the text that we hope improves it. Text marked in red below is added to the existing text:

In the abstract: "The nine contributing models prescribe a 50% increase in the cloud droplet number concentration (CDNC) of low clouds over the global oceans in an experiment called G4cdnc, with the purpose of counteracting the radiative forcing due to anthropogenic greenhouse gases under the RCP4.5 scenario."

At the end of page 2: "To attempt to alleviate this problem, the Geoengineering Model Intercomparison Project (GeoMIP) initiated a series of experiments where the models were to simulate MCB in a particular manner, with different degrees of complexity in the simulation design (Kravitz et al., 2013). The G1ocean-albedo experiment prescribes an increase in ocean albedo in the models at a rate intended to offset increasing global temperatures in response to a quadrupling of atmospheric $CO_2$ concentrations. The G4cdnc experiment investigated in this work simulates a 50 % increase in the CDNC of low marine clouds as described in more detail in Section 2. Finally, the G4sea-salt experiment involves an increase in sea salt emissions over tropical oceans at a rate intended to produce a radiative forcing of − 2.0 Wm$^{-2}$ under the RCP4.5 scenario. The three experiments have an increasing degree of complexity or realistic representation.»

*It would be useful to know how much of an overestimate applying the CDNC increase to all low clouds over the ocean may be to those that could realistically be subjected to a geoengineering scheme. What area of the earth does this correspond to in each of the different models? This is an important parameter to know: generating the requisite spray for MCB is no trivial task in terms of energy and scale-of-operation requirements.*

GeoMIP G4cdnc is an idealized experiment, which is not meant to realistically represent what could be done with marine cloud brightening. They were designed to test the robustness of models in simulating geographically heterogeneous radiative flux changes and to see their effects on climate. Increasing CDNC by 50% over all oceans is clearly an exaggeration of what could credibly be done. A more realistic GeoMIP marine cloud brightening experiment, G4sea-salt, is analyzed by Ahlm et al. (2017, ACP). The results identify the areas in the participating models where the sea salt seeding is the most effective at brightening the clouds. This includes the decks of persistent low level clouds off the west coasts of the major continents in the sub-tropics. A brief discussion around this is now

included at the very end of Section 5. As for calculating the ocean area that could realistically (energy-, scale-of-operation- and economical-wise) be subjected to MCB, compared to the area with low cloud amounts for the different models, we do not believe that we have the right data to quantify this in a way that is defensible.

*In the introduction there is reference to the finding of Alterskjaer et al., that seeding may lead to warming under certain conditions. Connolly et al. (2014, Phil Trans) discuss a similar finding in some detail. In short, the finding is that the Abdul Razzak et al parametrization does not work outside of the region they were originally tested and, in some ranges of parameter space, increasing the aerosol number concentration can lead to a spurious reduction in droplet number concentration. I note from table 1 that some of the models used in this study employ the same parameterisation, including NorESM. While CCN activation schemes are not a focus of this particular paper, some of the reasons for the effects talked about in the introduction may be due to CCN activation schemes, so it is an important point for other people working in this area.*

Thank you for pointing us towards the Connolly et al paper. This is an interesting point, and we agree it deserves mentioning. We have now added a sentence in the Introduction, referring to the Connolly et al. paper and pointing to the importance of the models' activation schemes. We have also added a small paragraph on this topic (second last paragraph) in Section 5. That being said, in this experiment, models did not increase their aerosol number concentrations, but rather increased their cloud droplet number concentrations directly, essentially bypassing this issue.

In the introduction: However, comparison between studies has been difficult partly due to different experimental design. Also, in studies based on one or only a few models, results will be very dependent on the particular models' parameterizations. For instance, Connolly et al. (2014) suggested that the unintended warming found in Alterskjær and Kristjánsson (2013) for seeding of some particle sizes (as mentioned above) was likely to be an artifact of the cloud parameterization scheme used in the model.

In Section 5: As seen in Table 1, there are large differences between models as to how the aerosol-cloud processes are parameterized, and this presumably has an impact on the MCB climate response. For instance, Morrison et al. (2009) found that the complexity of the cloud schemes had large impacts on stratiform precipitation; Penner et al. (2006) compared several models and concluded that the method of parameterization of CDNC can have a large impact on the calculation of the first indirect effect. CSIRO-Mk3L-1-2 and MPI-ESM-LR are the only models that have prescribed CDNC levels in their simulations, and these models stand out as having ERFs well above any of the other models. While an evaluation of which liquid cloud parameterizations are more appropriate is beyond the scope of this paper, this might be an indication that prescribing CDNC levels can lead to exaggerated responses to marine cloud brightening.

*I was not familiar with the Gregory regression method (section 3) until I did a search on the Internet. I think the original source should be cited. Would it be more complete to show / present the correlation coefficients associated with the analysis, to allow the reader to assess its suitability?*

In Section 2.2, we already state that "To estimate the effective radiative forcing of increasing CDNC by 50% in the different models, we use the method of Gregory et al. (2004), whereby the top of atmosphere radiative flux imbalance is regressed against the global annual average surface air temperature change compared to the RCP4.5 simulations."

To remind the reader of this explanation, we now refer back to this section when the ERF results are presented. We have also added an extra sentence after the above extract from Section 2.2, to further detail how this Gregory regression was performed;

To put the most weight on the initial changes, we use annually averaged values for years 2020-2030, but decadal mean values for years 2031-2070.

Regarding correlation coefficients between change in temperature and TOA radiative flux imbalance, we have now added a column in Table S1 showing the coefficients for each model as well as

indications of statistical significance. This table is now referred to in the results, in the section where the ERF are presented. Unsurprisingly, the models with the weakest ERF have the weakest correlations.

| | Correlation between surf. temp. change and TOA rad. flux imbalance | Correlation between surf. temp. change and baseline low cloud fraction | Correlation between surf. temp. change and baseline LWP for oceans |
|---|---|---|---|
| | Global mean correlation | Spatial correlation | Spatial correlation |
| BNU-ESM | -0.60 | -0.59 | -0.49 |
| CanESM2 | -0.83 | -0.09 | +0.43 |
| CSIRO-Mk3L-1-2 | -0.92 | -0.18 | +0.44 |
| GISS-E2-R | -0.26* | +0.08 | +0.04 |
| HadGEM2-ES | -0.69 | +0.06 | +0.50 |
| IPSL-CM5A-LR | -0.35* | +0.03 | +0.27 |
| MIROC-ESM | -0.56 | -0.49 | +0.47 |
| MPI-ESM-LR | -0.85 | +0.16 | +0.20 |
| NorESM1-M | -0.29* | -0.51 | -0.32 |
| **Median** | -0.55 | -0.13 | +0.45 |

**Table S1:** For the global mean correlations (leftmost column), globally and annually averaged temperature changes over the years 2020-2060 are correlated against corresponding changes in TOA radiative flux imbalance. For the spatial correlations (two rightmost columns), global matrices of annual mean changes (average of years 2020-2060) are correlated against baseline (20 first year of RCP4.5) values. All correlations except those marked by an asterisk are statistically significant at the 95% level. The lowest row shows correlation between model-median quantities (not the median of the individual model values above).

*Fig 4a was not referred to directly in the text.*
Figure 4a was referred to in the first paragraph of Section 4 (previously 3): "The model median geographical pattern of this radiative perturbation is shown in Fig. 4a."

*Although the models in the paper are important tools for climate assessment, the major issue with this kind of analysis is that the processes under investigation are not resolved at high enough grid resolution. There should be some caveat in the paper so that readers who are unfamiliar with the details are not mis-lead.*
We absolutely agree that in spite of all the advantages of using Earth system models in this type of study, this is a clear disadvantage. We have added the following paragraph to the end of Section 4, covering the reviewer's comments on both model resolution and overestimation by seeding clouds above all oceans:

A caveat of the present study is that the spatial resolution of global climate models is too coarse to resolve important processes such as convection or precipitation formation. These processes are instead parametrized, which may lead to unknown artifacts in the responses, dependent on the specific formulations used (see e.g. Clark et al. (2009) who compared precipitation forecast skills for convection-allowing and convective-parametrized ensembles). It is also important to point out that the G4cdnc experiment was not designed to give a realistic representation of the magnitude of cooling and other climate responses to MCB, but rather to test the robustness of models in simulating geographically heterogeneous radiative flux changes and to see their effects on climate. Increasing CDNC by 50% over all oceans is clearly an exaggeration of what could credibly be done. A more realistic GeoMIP marine cloud brightening experiment, G4sea-salt, is analyzed by Ahlm et al. (2017). Their results identify the areas in the participating models where the sea salt seeding is the most effective at brightening the clouds. This includes the decks of persistent low level clouds off the west coasts of the major continents in the sub-tropics. Alterskjær et al. (2012) investigated the susceptibility of

marine clouds to MCB based on satellite as well as model data, and reached a similar conclusion; large regions between 30°S and 30°N, and especially clean regions in the Pacific and Indian oceans, along with regions in the western Atlantic, are susceptible.

*The statement: "Our results suggest that liquid cloud parameterizations ought to be of appropriate complexity in order to attempt to model marine cloud brightening and the climate response." I did not see how this conclusion was arrived at. How do you know that liquid cloud parameterisations are complex enough?*
The reviewer is of course right – we have not performed any evaluation of different parameterizations. We have now modified our language from "our results suggest" to "we hypothesize";

We hypothesize that liquid cloud parameterizations ought to be of appropriate complexity in order to attempt to model marine cloud brightening and the climate response.

---

## Referee Report (RR1)

**Reviewe of revised verion:** *"Response to marine cloud brightening in a multi-model ensemble"*

The scientific clarity has been significantly improved in the revised version. In particular, additional supplementary figures and tables are helped greatly improving the manuscript. However minor revision is needed especially in "Post-processing and analyses section". I can therefore now recommend the paper to be published at ACP following only minor comments.

**Minor comments:**
- In the Post-processing analyses section, the years are confusing, for example, page 4 L6-7': In the above section, it is mentioned that, G4cdnc experiment is run for 50 Years, (2020-2069), after which cloud brightening experiment is terminated. However, here it is mentioned that, "we base our analyses on years 2020-2060" Please explain, why the base years are not 2020-2069? Again in the same section page 4 L16-17, The annual average is also confusing, more explanation is needed.
- Page 3. L33-34: "Note also that in BNU-ESM, the 50% increase in CDNC had to be obtained through a direct alteration of liquid droplet size, which means that its experiment set-up is slightly different from the other models", More information regarding liquid cloud droplet modification could be worthwhile.
- Page1. L39: 2015 21st, Please add comma.
- page6. L30: expand ITCZ.
- Page9. L4: add space between "ofCDNC".
- Figure 3b, Please include in the figure caption, "Dashed line indicate the termination period".
- Please make use of the abbreviations throughout the manuscript, for example LWP.

---

## Author Response (AR2)

**Response to reviewers**

We are happy that the reviewer feels that the manuscript is improved after the previous revisions, including the additional figures and tables in the supplement.

**Reviewer #1**
* * *
*• In the Post-processing analyses section, the years are confusing, for example, page 4 L6-7': In the above section, it is mentioned that, G4cdnc experiment is run for 50 Years, (2020-2069), after which cloud brightening experiment is terminated. However, here it is mentioned that, "we base our analyses on years 2020-2060" Please explain, why the base years are not 2020- 2069?*

We appreciate that the reviewer points out this discrepancy in our choice of analyses years. The choice to end the data period in 2060 stems from an early point in our analyses where data was lacking in the years after 2060 from some of the few models we had achieved data from. This changed, but in the process we forgot to change the analysis period. This has now been done consistently throughout the manuscript, so that the years 2020-2069 as used as a base for calculating changes. The results do not change at all qualitatively, and quantitatively only very little (the values of change in Table 3 do not change much). For instance, the model-median temperature reduction changes from -0.95 to -0.96 K. New figures are now used in the tables, and in the text.

*Again in the same section page 4 L16-17, The annual average is also confusing, more explanation is needed.*

We agree that our choice to use annual values in the first part of the time series, and decadal values in the latter, when performing the Gregory regressions, was not well explained in the text. We have now elaborated and clarified the text to read:
"To estimate the effective radiative forcing of increasing CDNC by 50% in the different models, we use the method of Gregory et al. (2004), whereby the global mean top of atmosphere radiative flux imbalance is regressed against the globally averaged surface air temperature change compared to the RCP4.5 simulations. Because of the time scales of the adjustments to the applied forcing, we follow Williams et al. (2008) and Gregory et al. (2004), and use annual means for the first decade, and decadal means thereafter; the period of 2030 - 2069. This allows more weight to the years when the rapid adjustments dominate, before the slow feedbacks have more impact in the later part of the simulation".

The following sentence has also been added to the caption of Fig. 3:
"Each box represent global annual mean for each of the first ten years and decadal means for the remaining part of the simulations (i.e. last four decades, 2030 - 2069)."

*• Page 3. L33-34: "Note also that in BNU-ESM, the 50% increase in CDNC had to be obtained through a direct alteration of liquid droplet size, which means that its experiment set-up is slightly different from the other models", More information regarding liquid cloud droplet modification could be worthwhile.*

The different model set-up of BNU-ESM did indeed merit a bit more explanation. We have now changed the last sentence of section 2.1 to the following:

"Note also that for BNU-ESM, the model design precluded a direct change in the CDNC. Therefore, instead of multiplying 50% increase in CDNC by 1.5 to obtain the 50% increase, they had to approximate be obtained the cloud seeding through a direct alteration of liquid droplet sizes (droplet radii are multiplied by the factor $1.5^{1/3}$), which means that its experiment set-up is slightly different from the other models."

*• Page1. L39: 2015 21st, Please add comma.*

The comma is now added.

*• page6. L30: expand ITCZ.*

ITCZ is expanded as requested.

*• Page9. L4: add space between "ofCDNC".*

We were not able to find this error, but have at least checked and made sure that "ofCDNC" does not occur anywhere in the manuscript.

*• Figure 3b, Please include in the figure caption, "Dashed line indicate the termination period".*

We thank the reviewer for this suggestion – we chose to add the following text to the caption "Dotted vertical line indicates the onset of the termination period".

*• Please make use of the abbreviations throughout the manuscript, for example LWP.*

We have done as the reviewer requests, and the manuscript is now more consistent in its use of abbreviations, such as LWP, TOA and CDNC.